# In Silico Characterization of the Secretome of the Fungal Pathogen *Thielaviopsis punctulata*, the Causal Agent of Date Palm Black Scorch Disease

**DOI:** 10.3390/jof9030303

**Published:** 2023-02-27

**Authors:** Biju Vadakkemukadiyil Chellappan, Sherif Mohamed El-Ganainy, Hind Salih Alrajeh, Hashem Al-Sheikh

**Affiliations:** 1Department of Biological Sciences, College of Science, King Faisal University, P.O. Box 420, Al-Ahsa 31982, Saudi Arabia; 2Department of Arid Land Agriculture, College of Agriculture and Food Sciences, King Faisal University, P.O. Box 420, Al-Ahsa 31982, Saudi Arabia; 3Agricultural Research Center, Plant Pathology Research Institute, Giza 12619, Egypt

**Keywords:** fungal pathogen, secretory proteins, CAZymes, effectors

## Abstract

The black scorch disease of date palm caused by *Thielaviopsis punctulata* is a serious threat to the cultivation and productivity of date palm in Arabian Peninsula. The virulence factors that contribute to pathogenicity of *T. punctulata* have not been identified yet. In the present study, using bioinformatics approach, secretory proteins of *T. punctulata* were identified and functionally characterized. A total of 197 putative secretory proteins were identified, of which 74 were identified as enzymes for carbohydrate degradation (CAZymes), 25 were proteases, and 47 were predicted as putative effectors. Within the CAZymes, 50 cell wall-degrading enzymes, potentially to degrade cell wall components such as cellulose, hemicellulose, lignin, and pectin, were identified. Of the 47 putative effectors, 34 possessed at least one functional domain. The secretome of *T. punctulata* was compared to the predicted secretome of five closely related species (*T. musarum*, *T. ethacetica*, *T. euricoi*, *T. cerberus*, and *T. populi*) and identified species specific CAZymes and putative effector genes in *T. punctulata*, providing a valuable resource for the research aimed at understanding the molecular mechanism underlying the pathogenicity of *T. punctulata* on Date palm.

## 1. Introduction

Fungal pathogens cause huge yield losses in agricultural crops and post-harvest products worldwide [1]. According to the Food and Agriculture Organization (FAO), an estimated $220 billion global economy is lost due to fungal disease every year. To prevent such losses, farmers use several fungicides, which is not only an ineffective method as the pathogens gain resistance against these chemicals quickly but is also very harmful to humans and the environment. Alternatively, genetic approaches, including the use of resistance genes, are considered safer and more durable. However, the selection pressure imposed by single resistance genes in host plants can force rapid evolutionary changes in pathogens that often lead to resistance breakdown. For example, *Fusarium oxysporum* f. sp. *lycopersici*, a wilt pathogen of tomato, has evolved multiple times as different races to evade host resistance when a cultivar with a new resistance gene have been introduced in the field [2]. Therefore, to achieve more durable resistance against fungal pathogens, a deep understanding of the virulence factors secreted by the pathogen and the resulting plant immune responses is inevitable [3].

For the successful penetration and colonization, pathogens have to overcome multiple layers of plant immunity [4]. The first layer of immunity in plants is triggered by pattern recognition receptors, which recognizes pathogen-associated molecular patterns, for example, chitin in fungal pathogens. This layer of immunity is called PAMP-triggered immunity (PTI) [5]. Although PTI is effective against a broad spectrum of microorganisms, pathogens overcome PTI by secreting so-called effector proteins that manipulate cellular processes in the host to facilitate effector-triggered susceptibility (ETS) [6,7,8,9,10]. In turn, plants have evolved a second layer of immunity in which they employ another type of receptor called resistance (R) proteins [11]. R proteins recognize specific pathogen effectors or their effects on the plant cell, resulting in effector-triggered immunity (ETI) [12]. Effector proteins in the pathogen that are recognized by specific R proteins in the host are called avirulence proteins (Avr) [13]. The interaction between an R protein and its cognate Avr protein leads to a disease resistance response, often a so-called hypersensitive response (HR), a programmed cell death at the site of infection site by which further growth of the pathogen in the plant is restrained [14,15,16]. In response to this, pathogens may overcome ETI by loss-of-function of the avirulence protein or by employing new virulence factors. In the plant, new R proteins may evolve that recognize other pathogen effectors, which often leads to a molecular arm race between the pathogen and its host plant [17,18].

Several plant–microbe interaction studies have shown that effectors play major roles in determining pathogenicity of many phytopathogens [6,19]. Effectors are proteins secreted by pathogens into the extracellular and intracellular spaces of host plants to manipulate host targets. These proteins are approximately 50–300 amino acid residues in length, containing an N-terminal signal peptide with a highly specific sequence, with no transmembrane structural domain, no anchor site for glycosylphosphatidylinositol (GPI), no subcellular localization signal for mitochondria or other intracellular organelles, and being rich in cysteine residues [8]. These typical characteristics enable scientists to predict effectors from many pathogens’ genomes. For example, in a recent study, the draft genome was used to predict the putative effectors of *Fusarium oxysporum* f. sp. *albedinis*, a pathogen that causes dieback disease on date palms [20].

Black scorch disease caused by the fungus *Thielaviopsis punctulata* is an important problem confronting the date palm industry, with losses of >50% in newly planted offshoots and fruit [21]. This disease has been reported on date palm in many date-growing areas in the world, including Saudi Arabia, Oman, Qatar, United Arab Emirates, Spain, etc. [22,23,24,25,26,27]. Once they have penetrated any vegetative part of the plant, this fungus causes severe rotting to occur in the buds, heart, inflorescence, leaves, and/or trunk of the plant. The application of fungicide such as difenoconazole is an effective control against black scorch disease in date palm plants [27]. In addition, traditional horticultural practices such as avoidance of wounds of trees and the removal and burning of diseased plants also helped to reduce the spread of disease. The use of biocontrol agent is also another method to compact this disease [28,29]. However, as a part of the long-term disease management approach, more recent genome-based molecular biology and biotechnological research can provide fair control and can target fungal pathogens in date palm.

Although a draft genome of *T. punctulata* has been published, its secretory proteins have not been extensively characterized yet [30]. In this study, using a bioinformatics approach, we comprehensively annotated secretory proteins in *T. punctulata* genome. This research provides valuable resource on the systematic analysis of the *T. punctulata* cell wall-degrading enzymes, proteases, pathogenicity-related proteins, and putative effector proteins and will be a valuable resource for the future *T. punctulata*-date palm molecular interaction studies.

## 2. Materials and Methods

### 2.1. Sequence Information and Gene Prediction

The gene models of the draft genome sequence (NCBI accession: GCA_000968615.1) of the *Thielaviopsis punctulata* isolate CR-DP1_NODE_1 was used for the prediction of secretome. For the comparative analysis, the gene models were predicted for the closely related species, viz; *Thielaviopsis musarum* (NCBI accession: GCA_001513885.), *Thielaviopsis cerberus* (GCA_016859225.1), *Thielaviopsis ethacetica* (NCBI accession: GCA_001599055.1), *Thielaviopsis euricoi* (NCBI accession: GCA_001599615.1), *Thielaviopsis populi* (NCBI accession: GCA_017591655.1). The gene models were predicted according to the method used by Wingfield et al. [30]. To infer the phylogenetic relationship among Thielaviopsis species, 50 shared orthologs were selected randomly and a concatenated alignment was made. The relationship was constructed by MEGA11 using the Maximum Likelihood method and JTT matrix-based model (based on 1000 bootstrap replications).

### 2.2. Prediction of the Secretome

We used a pipeline described previously to predict fungal secretome [31,32]. Briefly, SignalP (version 6.0) was used in combination with Phobius server [33,34]. The sequences, that were predicted to carry a signal peptide by both programs were selected for further screening. To exclude the transmembrane proteins, DeepTMHMM server was used [35]. The endoplasmic reticulum-targeting protein sequences were removed by scanning the sequences for PS00014 ER motif retention against the Prosite database with the ScanProsite web server [36]. The subcellular localization of the proteins was predicted using TargetP and WoLF PSORT servers [37,38]. The proteins harboring glycophosphatidylinositol anchor motifs were predicted using NetGPI (version 1.1) [39].

### 2.3. Annotation of Secretory Proteins

The refined secretome were scanned against Uniprot, PFAM, InterPro, and Gene3D to retrieve functional annotation of the predicted proteins [40,41,42,43]. The CAZy database and dbCAN web server were used to retrieve the annotation of carbohydrate-degrading enzymes [44,45]. For the effector prediction, the standalone software, EffectorP (version 3.0), combined with the manual inspection was used [46]. In addition, the BlASTP (E value < 1 × 10^−10^) was used to search against the pathogen–host interaction database (PHI database) to find similarities to known effectors and virulence factors [47]. Proteolytic enzymes were identified using a BlastP search against MEROPS database.

## 3. Results and Discussion

### 3.1. Prediction of Gene Models and Orthologue Analysis

The draft genome sequence of the *Thielaviopsis punctulata* (NCBI accession: GCA_000968615.1) and the 5296 predicted gene models were used for the identification of secretome. In addition, for the comparative analysis, we have also predicted the genes and the encoded proteins for 5 closely related species of *T. punctulata* viz—*T. musarum*, *T. ethacetica*, *T. euricoi*, *T. populi*, and *T. cerberus*—by utilizing the draft genome of these species available publicly (Table 1) [30]. The gene models were predicted according to the method described previously [30]. The number of predicted proteins in each species is given in Table 1. The gene models predicted for the closely related species were given in the Appendix A. The orthologue analysis using the whole proteome revealed that all six species form 7001 clusters, 3719 orthologous clusters (at least contains two species) and 3282 single-copy gene clusters (Figure 1, Appendix A). The number of singletons (The proteins which did not form any cluster) vary among the species. The highest number of singletons were found for *T. cerberus* (354), whereas the lowest were found for *T. punctulata* (50), which suggested that 94.4% of predicted proteins of *T. punctulata* had orthologue in other species. A phylogenetic tree was constructed using a concatenated alignment of 50 single copy orthologue proteins and revealed that *T. cerberus* was closely related to *T. punctulata*. The close genetic similarity between *T. punctulata* and *T. cerberus* was shown previously using internal transcribed spacer (ITS), β-tubulin, and transcription elongation factor 1-α DNA markers (Figure 1B) [25].

### 3.2. Secretome Identification and Analysis

The methodology used to predict the secretome of *T. punctulata* is illustrated in Figure 2. Using a combination of SignalP and Phobius server, of the 5296 total proteins, 314 proteins were predicted to have a signal peptide at their N-terminal region. Among these 314 proteins, 63 transmembrane proteins were excluded, and the remaining 251 proteins were scanned for an endoplasmic reticulum (ER)-targeting signal to exclude the proteins that remain in the endoplasmic reticulum. Of the 251 proteins, 13 were predicted to have PS00014 ER motif and were excluded from further analysis. The remaining 238 proteins were predicted as “extra-cellularly localized” through TargetP and WoLF PSORT analysis. Next, of the 238 proteins, 41 proteins were predicted to harbor glycophosphatidylinositol anchor motifs using NetGPI (version 1.1), which likely represent surface proteins rather than secreted effectors and were excluded. This resulted in a list of 197 “refined secretome”, which is 3.7% of the whole predicted proteome of *T. punctulata* (Table 1). Using this method, the secretome of *T. musarum*, *T. ethacetica*, *T. euricoi*, *T. populi*, and *T. cerberus* were also predicted. Overall, the number of “refined” secretome in all six species ranged from 150 (*T. cerberus*) to 215 (*T. ethacetica*) (Table 1, Figure 2B).

### 3.3. Structural and Functional Characterization of Secretome

The length of 197 refined secretome of *T. punctulata* ranged from 78 aa to 1356 aa. Of these, 43% (86) of proteins had a length of 78 aa to 399 aa, which indicated that small secretory proteins were enriched in the secretome of *T. punctulata* (Figure 2C). 

Moreover, the molecular weight (MW) of secretory proteins ranged from 8.0 kDa to 147 kDa, and for most of the secretory proteins, it ranged between 10 and 49 kDa (56%) (Figure 2D). Similarly, the theoretical isoelectric point (pI) of the secretory proteins ranged from 3.8 to 9.4, of which the majorities (>77) pI ranged from 4–5.9 (Figure 2E). A similar pattern of length, PI and MW distribution, was also observed in closely-related species of *T. punctulata* (Figure 2C–E). The refined secretome of *T. punctulata* was characterized based on their matches in Uniprot, NCBI fungal reference proteome, Interpro and PFAM, Gene3D. The domain analysis revealed the presence of at least one function domain in 162 proteins. The most represented domains were Peptidase_S8 (PF00082), Auxiliary Activity family 9 (PF03443), Egh16-like (PF11327), Glycosyl hydrolases family 16 (PF00722), Glycosyl hydrolases family 43 (PF04616), Asp (PF00026), etc. (Figure 3A). Based on the sequence homology, gene ontology (GO) terms were assigned to 132 proteins which were further grouped into three major functional categories: biological process (71 proteins), molecular function (116 proteins), and cellular components (39 proteins) (Figure 3B). The biological processes include carbohydrate metabolic process (GO:0005975), polysaccharide catabolic process (GO:0000272), lipid metabolic process (GO:0006629), cellulose catabolic process (GO:0030245), chitin catabolic process (GO:0006032), arabinan catabolic process (GO:0031222), cellular aromatic compound metabolic process (GO:0006725), and xylan catabolic process (GO:0045493) (Figure 3B). The prominent category under the molecular function includes hydrolase activity (GO:0004553), serine-type endopeptidase activity (GO:0004252), cellulose binding (GO:0030248), endo-1,4-beta-xylanase activity (GO:0031176), oxidoreductase activity (GO:0016614), etc. (Figure 3B). The cellular component includes extracellular region (GO:0005576), cell wall (GO:0005618), and membrane (GO:0016020) (Figure 3B).

### 3.4. Carbohydrate Active Enzymes

Carbohydrate active enzymes, also called CAZymes, are a general group of enzymes involved in the biosynthesis and breakdown of carbohydrate and glycoconjugates. They are categorized into glycoside hydrolases (GH), polysaccharide lyases (PL), carbohydrate esterases (CE), auxiliary activity (AA), glycosyltransferasse (GT), and carbohydrate-binding module (CBM) classes [48,49]. To identify the CAZymes in *T. punctulata* and its closely related species, different sources of information such as blast description, Gene ontology, EC number, PFAM domain, and the results of annotation with CAZy database were combined [48]. Of the 197 refined secretome of *T. punctulata*, 75 proteins were identified as putative CAZymes, including 47 GH, 19 AA, 4 PL, 4 CE, and 1 GT (Figure 4, Table 2). Of these 75 CAZymes, 31 proteins contain multiple CAZymes modules. Among these, eight proteins possess two or more copies of the same CAZymes module that include two proteins with four copies of AA5 module and six proteins with two copies of GH3, GH7, GH20, GH32, and PL modules, respectively. Overall, 24 proteins contained two or three CA-Zyme modules of different types. In addition, 5 GH, 1 PL, 1 AA9, and 1 AA3 protein also contained a CBM module. CAZymes were also identified from five other species and found more numbers in *T. punctulata* (Table 2).

Except for the GT family, all other families of CAZymes (GH, CE, PL, and AA) were considered as cell wall-degrading enzymes since they are involved in the breakdown of plant cell wall components such as cellulose, hemi cellulose, pectin, and lignin [50,51]. Cellulose is an organic polysaccharide composed of a linear chain of hundreds of β-linked D-glucose units, and the enzymes involved in the breakdown of cellulose are exo-β-1,4-glucanases, endo-β-1,4-glucanases, β-1,4-glucosidases, cellobiose dehydrogenase, and lytic cellulose monooxygenase [51,52,53,54]. Based on the substrate specificity, of the 75 CAZymes of *T. punctulata*, 20 were predicted to be involved in the degradation of cellulose including seven endo-β-1,4-glucanases, one exo-β-1,4-glucanase (cellulose 1,4-beta-cellobiosidase (reducing end)), two β-glucosidase, two cellobiose dehydrogenase, and five lytic cellulose monooxygenase (Table 2). The CAZymes families containing endo-β-1,4-glucanases include GH5 (5), GH45 (1), and GH7 (1). One exo-β-1,4-glucanase was found in the GH7 family and five β-glucosidase were found in GH3 (3) and GH131 (2) families, respectively (Table 2). Seven lytic cellulose monooxygenases (EC:1.14.99.54) were found in AA9 and two cellobiose dehydrogenase (EC:1.1.99.18) in AA3 families (Table 2). 

Hemicellulose is another major component of the plant cell wall, which include xyloglucans, xylans, mannans and glucomannans, and beta-(1-->3,1-->4)-glucans. The major enzymes involved in hemicellulose degradation are L-arabinanases, D-galactanases, D-mannanases, and D-xylanases [55]. Apart from this, Endo-_-1,4-glucanases with xyloglucanase activity were also identified from several fungal species [51]. Of the 74 CAZymes, 16 proteins were predicted to be involved in the degradation of hemicellulose, which included 14 GH and 2 CE. The GH group included 4 GH43, 2 GH10, 2 GH11, one member from GH38, GH51, GH53, GH93, GH115, and GH125, respectively (Table 2). The GH43 family consists of 3 exo-β-1,3-galactanases (EC:3.2.1.145) and one endo-α-1,5-L-arabinanase (3.2.1.99), which were predicted to be involved in the cleavage of galactans and arabinans, respectively (Table 2) [55]. Both the GH10 and GH11 family encode endo-1,4-β-xylanase (EC:3.2.1.8), which catalyzes endohydrolysis of (1->4)-beta-D-xylosidic linkages in xylans (Table 2) [55]. The GH38, GH51, GH53, GH93, GH115, and GH125 families encode α-mannosidase (involved in the cleavage of mannose), α-L-arabinofuranosidase (involved in the cleavage of arabinans), endo-β-1,4-galactanase (involved in the cleavage of galactans), exo-α-L-1,5-arabinanase (involved in the cleavage of arabinans), xylan α-1,2-glucuronidase (involved in the cleavage of xylans), and exo-α-1,6-mannosidase (involved in the cleavage of mannans), respectively (Table 2) [55]. The two CE members belong to CE5 family encoding acetyl xylan esterase (3.1.1.72), which catalyzes the hydrolysis of acetyl groups from polymeric xylan (Table 2) [56].

Within the refined secretome, members of polysaccharide lyases (PLs), including two pectin lyase (EC:4.2.2.10), one pectate lyase (EC:4.2.2.2), one rhamnogalacturonan endolyase (EC:4.2.2.23), and one GH28 polygalacturonase, were also identified (Table 2). These enzymes are known to degrade pectin (Table 1) [57]. In addition to glycoside hydrolases, members of Auxillary activity (AA) families were identified with the potential to degrade lignin (Table 2) [57]. These include three AA1 laccases (EC:1.10.3.2), one AA2 peroxidase (EC 1.11.1.-), one AA2 versatile peroxidase (EC:1.11.1.16), one aryl alcohol oxidase (EC:1.1.3.7), and one AA5 Oxidase with oxygen as acceptor (EC:1.1.3.-) (Table 2). In addition to these cell wall-degrading enzymes, the refined secretome also contains two starch, one sucrose, one trehalose, three glucans, and three callose-degrading enzymes (Table 2). 

The comparison of CAZymes in closely-related species revealed 59 different classes of CAZymes in all species, including *T. punctulata* (Figure 4, Table 3, Appendix A). The secretome of all these species were rich in secreted CAZymes families, especially those involved in plant cell wall degradation (PCWD). High numbers of secreted CAZymes involved with PCWD have also been found in the genomes of several *Botryosphaeriaceae* pathogens [58]. However, the occurrence of these classes varied among *Thielaviopsis* species. For example, only 11 classes of CAZymes (AA1, AA5, GH10, GH11, GH16, GH17, GH20, GH32, GH43, GH76, and PL1) were found common in the secretome of all species (Figure 4, Table 3, Appendix A). In addition, some classes of CAZymes were found to be specific for some species, and notably 10 classes of CAZymes (GH64, GH125, GH45, GH132, AA16, GH128, GT4, GH51, GH115, GH38) were found only in the secretome of *T. punctulata* (Figure 4, Table 3, Appendix A), suggesting that they might have a species-specific role in black scorch disease of date palm.

### 3.5. Secreted Proteases

Several studies have shown that plant pathogenic fungi secrete proteases that degrade plant antimicrobial proteins and protease inhibitors (PIs) to facilitate virulence [59]. The BlastP search against MEROPS database resulted in the identification of 24 putative proteases from the 197 refined secretome, which were classified into several groups based on their catalytic residues (Appendix A). Among the proteases, serine proteases (13) were dominant, followed by metallo proteases (6), aspartic proteases (4), and carboxy protease (1). The serine proteases included S8, S10, and S28 families. Among these, S8 was found to be dominant (Appendix A). Members of metallo protease were further classified into M14, M28, M36, and M43 families based on their similarity to the known members from these families. Members of the same classes of proteases were also identified in the closely related species (Appendix A). Comparison to closely related species revealed that Serine protease S8 was prominent in *T. unctulate*, whereas S9 was completely absent (Figure 5).

### 3.6. Putative Effector Proteins

EffectorP, combined with the manual inspection, was employed to identify the putative effector proteins with the following characteristics: a signal peptide for secretion, no trans-membrane domains, fairly small size, and cysteine-rich [9,31,60]. This analysis resulted in the identification of 47 proteins as putative “effector” candidates (Figure 6). The length of these effector candidates varied from 78 to 392 Amino acids, of which 15 candidates were 100 to 200, 16 were 200 to 300, and 14 were 300 to 400 amino acids in length, respectively (Table 4). Two candidates (KKA29758.1 and KKA27537.1) were found with a length of less than 100 amino acids. The number of cysteine residues varied from 2 to 8 in the selected putative effectors, of which 70% were found with more than four cysteine residues (Table 4). Of the 47 putative effector proteins, functional domains were identified in 34 proteins, notably five putative effectors (KKA26926.1, KKA26947.1, KKA27553.1, KKA27672.1, KKA29410.1) possessed an Egh16-like (PF11327) domain. Three putative effectors (KKA26938.1, KKA27913.1, KKA28390.1) possessed copper bind domain (PF00127) (Figure 4, Table 4. A necrosis-inducing protein (NPP1) domain (PF05630) was identified in one protein (KKA27476.1). Seven putative effectors possessed a domain of unknown function (DUF) and 13 had no known functional domain in it (Figure 6, Table 4). Orthologue analysis found that *T. punctulata* shared 24 effectors with other closely related species, of which 3 proteins were shared with all closely related species (*T. musarum, T. euricoi, T. populi, T. ethacetica and T. cerberus*) (Figure 7). Two pairs of duplicated effector proteins were identified in the secretome of *T. punctulata* (KKA28270.1 and KKA26687.1 and KKA27979.1 and KKA29712.1). These paralogs showed 100% identity to each other, indicating a recent duplication. Interestingly, 18 effectors were found to be specific to *T. punctulata*, and functional analysis of these effectors may reveal more insight into the pathogenicity of *T, punctulata* on Date palm (Table 4).

### 3.7. Putative Virulence Factors

To identify the homologs of pathogenicity-associated genes in other phytopathogens, we screened 197 refined secretome, including all the putative effectors against the PHI (Pathogen–host interactions) database [47]. The protein sequences in the PHI database are classified into different categories such as loss of pathogenicity, reduced virulence, unaffected pathogenicity, increased virulence, effector (plant avirulence determinant), lethal, enhanced antagonism, resistant to chemical, and sensitivity to chemicals based on the results of mutation experiments. For example, the “Loss of pathogenicity” group includes proteins for which the mutant strains fail to cause diseases in host compared to the wild type. Based on the PHI annotation, of the 197 secretomes, 49 had PHI homologue, including 27 CAZymes, 7 proteases, and 15 putative effectors (Appendix A). Of the 27 CAZymes, four were assigned as effector (plant_avirulence_determinant), including three lytic cellulose monooxygenases in AA9 CAZymes family (KKA27328.1, KKA28497.1 and KKA25992.1) and one endo-β-1,4-xylanase in GH11 family (KKA30007.1). Both these enzymes were reported to contribute to the virulence of several plant pathogenic fungi. The homologue of lytic cellulose monooxygenase in *Magnaporthe oryzae* (MoCDIP) induced cell death when it was expressed in rice plant cells [61]. Similarly, a lytic cellulose monooxygenase gene (PHEC27213) in *Podosphaera xanthii* was shown to suppress the chitin-triggered immunity in cucurbit host [62]. Endo-β-1,4-xylanase was also shown to involved in the pathogenicity of many fungal pathogens, including *Verticillium dahlia*, *Ustilago maydis*, *Valsa mali*, etc. [63,64,65]. In addition, 13 CAZymes were also assigned as “reduced virulence” according PHI database, which includes three Pectin lyases (PL), two laccases (AA1), two β-glucosidases (GH03), one peroxidase (AA2), Oxidase (AA5), endo-1,4-β-xylanase (GH10), exo-α-1,6-mannosidase (GH25), and licheninase (GH16). The contribution of these genes in virulence was demonstrated for many plant-pathogenic fungi [66,67,68,69,70]. Of the CAZymes, six were assigned as “Unaffected category” (Appendix A). Among the 25 proteases identified, two metallo peptidases (KKA26166.1 and KKA27603.1) were assigned as “loss of pathogenicity” and one carboxy peptidase (KKA30601.1) was assigned as “reduced virulence” (Appendix A). The homologs of these two metallo peptidases were shown to require the pathogenicity of *Fusarium oxysporum* f. sp. *lycopersici* and *Magnaporthe oryzae* on their respective hosts [71,72]. In addition, two proteases were assigned as “unaffected pathogenicity” according to PHI annotation (Appendix A). From the putative effectors group, PHI partners were identified for 15 proteins, of which one was assigned as “loss of pathogenicity”, which encodes an acetylglucosaminyl phosphatidylinositol deacetylase (KKA29596.1) (Appendix A). The homologue of this protein in *Colletotrichum graminicola*, the causal agent of maize anthracnose, was shown to require cell-wall integrity and pathogenicity [73]. Thirteen effectors were assigned as “reduced virulence”, and homologs of these proteins were shown to be required for pathogenicity in many fungal species. Notably, the most enriched group of effectors in *T. punctulata* were Egh16-like proteins, and their homologue in PHI database (PHI:256) was shown to act as virulence factors in rice blast fungus *Magnaporthe grisea* [74]. The Egh16 family members were also shown to be involved in the virulence of many pathogenic filamentous fungi. For example, two Egh16-like factors in *Erysiphe pisi*, EpCSEP087 and EpCSP083, were found to be highly induced during early infection stages on pea, suggesting a critical role in appressorium penetration and pathogenesis [75].

## 4. Conclusions

In the current study, the potential secretory proteins of *Thielaviopsis punctulata* were extensively characterized using a well-designed bioinformatics approach. Within the secretome, 74 CAZymes, 25 proteases, and 47 putative effector proteins were identified. For the comparative analysis, the genes models for five closely related species of *T. punctulata* were predicted and the secretome of each species was also well-characterized. The comparative analysis revealed that all *Thielaviopsis* species studied possessed species-specific CAZymes families, putative effectors, and several putative virulence factors. The current study will be a valuable source for the studies aimed at understanding the pathogenicity mechanism not only in *T. punctulata*–date palm interaction, but also in other *Thielaviopsis* species and their respective hosts.

## Figures and Tables

**Figure 1 jof-09-00303-f001:**
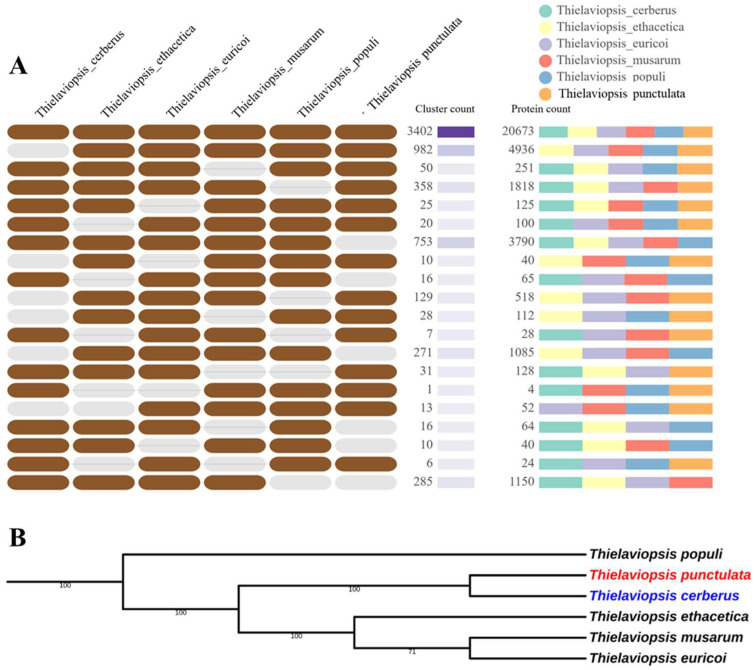
Orthologue gene clustering and phylogenetic analysis using predicted proteins of six *Thielaviopsis* species. (**A**). Orthologue clusters. Orthologue gene clusters were identified and visualized using the OrthoVenn2 web platform. The e-value cut off 1 × 10^−10^ was used for the analysis. (**B**). Phylogenetic tree of six *Thielaviopsis* species (*T. populi, T. ethacetica, T. cerberus, T. euricoi, T. musarum*, and *T. punctulata*) inferred from concatenated alignment of 50 shared orthologue proteins. The relationship was constructed by MEGA11 using the Maximum Likelihood method and JTT matrix-based model (based on 1000 bootstrap replications). The percentage of replicate trees in which the associated taxa clustered together in the bootstrap test are shown next to the branches.

**Figure 2 jof-09-00303-f002:**
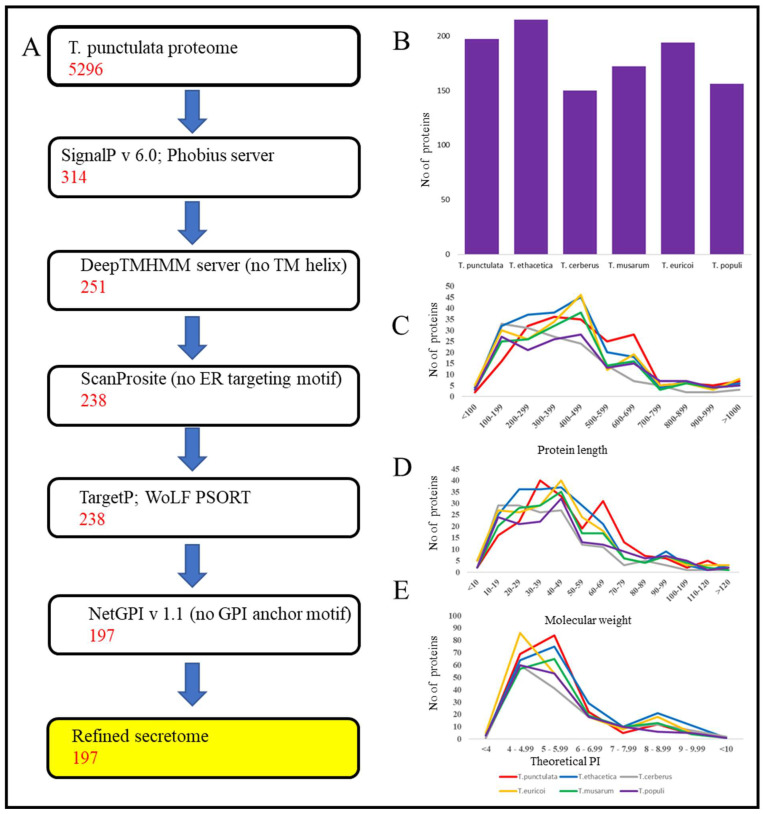
Secretome prediction in *Thielaviopsis punctulata* and its closely related species. (**A**) Pipeline for the prediction of secretome. The number of proteins in each step is given for *T. punctulata*. (**B**) Number of secretome of all six species analyzed. (**C**) Length distribution of the secretome. (**D**) Molecular weight of the secretome. (**E**) Theoretical PI of the secretome.

**Figure 3 jof-09-00303-f003:**
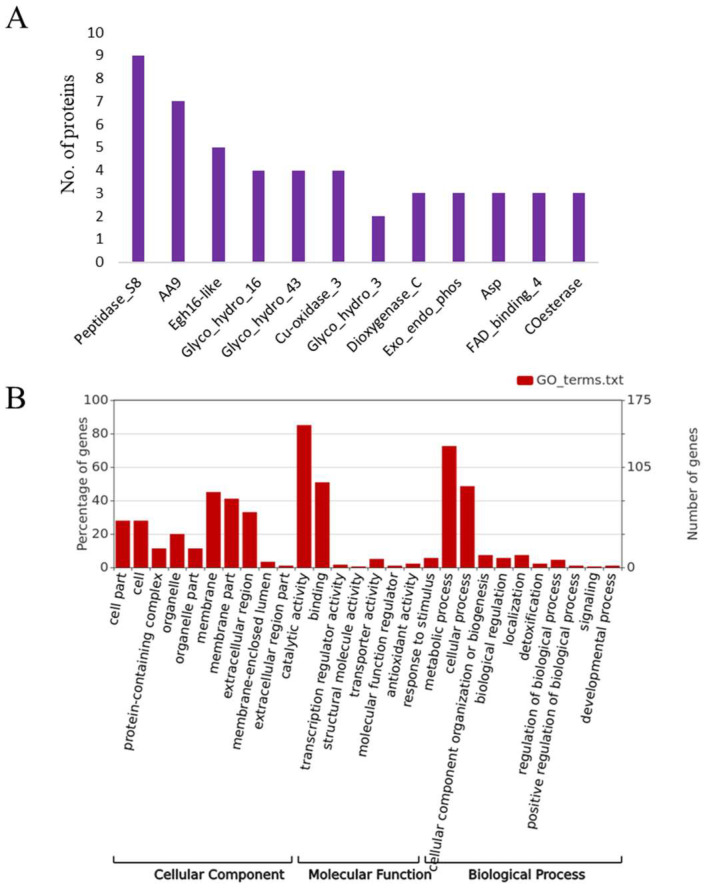
Functional domain and Gene ontology of secretome of *Thielaviopsis punctulata*. (**A**) Most enriched domains are shown in *x* axis, and *y* axis denotes the number proteins with respective domain. (**B**) Gene ontology annotation of genes based on domains present in the encoded proteins.

**Figure 4 jof-09-00303-f004:**
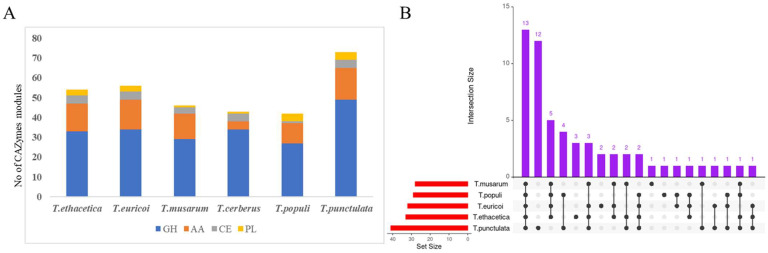
Overview of carbohydrate-activating enzymes (CAZymes) in *T. punctulata* and its closely related species. (**A**) CAZymes modules in *T. punctulata* and its closely related species. *y* axis denotes number different CAZymes modules. (**B**) Number of CAZY families shared between *T. punctulata* and its closely related species. Purple bar indicates the intersection. Red bar indicates the data size. Dots and lines connect the species with shared orthologues.

**Figure 5 jof-09-00303-f005:**
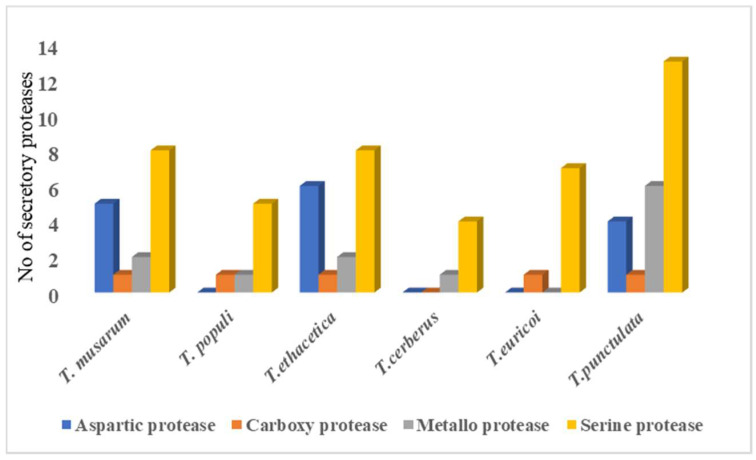
Secretory proteases in *T. punctulata* and its closely related species.

**Figure 6 jof-09-00303-f006:**
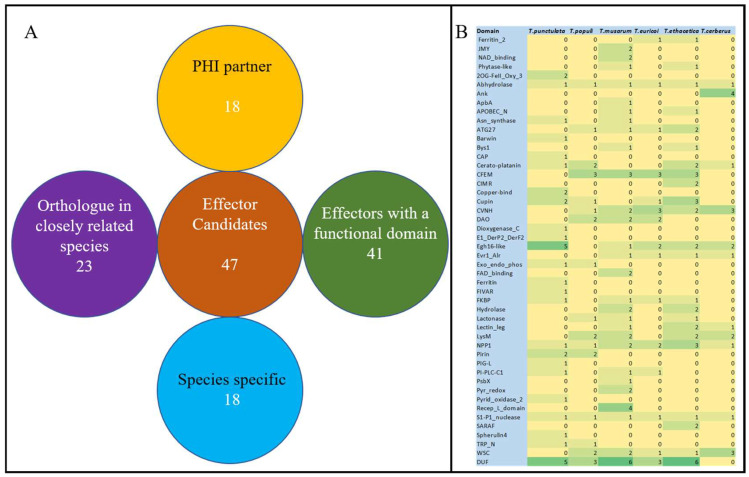
Putative effector proteins in the secretome of *Thielaviopsis punctulata*. (**A**) Overview: EffectorP identified a total of 47 putative effectors. Among these, functional domains were identified in 41 effectors. 18 putative effectors were identified as having a PHI partner, orthologues for 23 were found in the closely related species, and 18 were identified as species-specific. (**B**) Functional domains in the putative effector proteins of *T. punctulata* and closely related species. The number of domains of range 0–6 is represented by a golden to green color gradient.

**Figure 7 jof-09-00303-f007:**
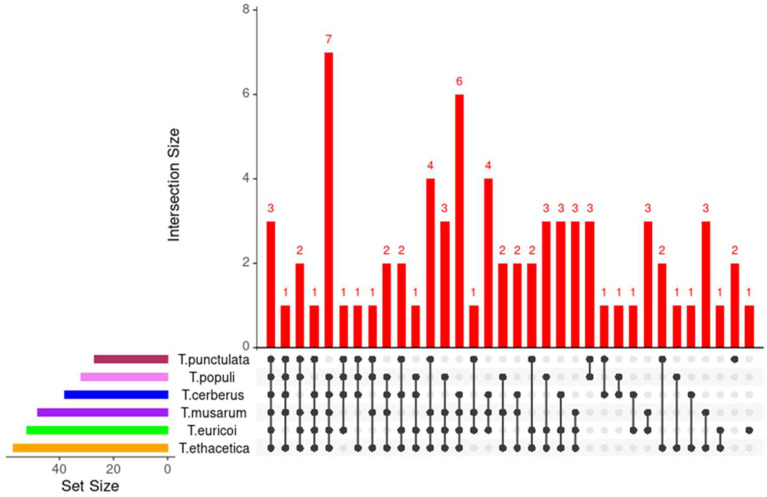
Orthologues of putative effector proteins in *Thielaviopsis* species. The number of orthologue genes shared by *T. punctulata* and its closely related species is shown as a red bar. The size of clusters in species is represented by different colors. Dots and lines connect the species with shared orthologues. The figure is generated by UpsetR.

**Table 1 jof-09-00303-t001:** Overview of Secretome of *T. punctulata* and its closely related species.

Species	Genome Size (Mb)	Total Proteins	Refined Secretome	% of Secretome	Cazymes	Proteases	Putative Effectors
*T. punctulata*	28.1	5296 ^a^	197	3.71	75	24	47
*T. ethacetica*	29.4	7079 ^b^	215	3.03	54	17	68
*T. cerberus*	28.6	5591 ^b^	150	2.68	43	5	59
*T. musarum*	28.4	6801 ^b^	172	2.52	46	16	56
*T. euricoi*	29.6	7004 ^b^	194	2.76	56	8	62
*T. populi*	23.9	6220 ^b^	156	2.50	42	7	55

a—NCBI accession: GCA_000968615.1, b—gene prediction in this study.

**Table 2 jof-09-00303-t002:** List of CAZymes in the secretome *T. punctulata*.

CAZy Family	Protein Id	PFAM Id	E.C. Number	Enzyme Name	Substrate
AA1	KKA28993.1	PF07731.17	1.10.3.2	Laccase	lignin
	KKA29108.1	PF07732.18	1.10.3.2	Laccase	lignin
	KKA30055.1	PF07732.18	1.10.3.2	Laccase	lignin
AA16	KKA29756.1	PF03067.18	na	Lytic cellulose monooxygenase	cellulose
AA2	KKA28039.1	PF00141.26	1.11.1.16	versatile peroxidase	lignin
	KKA29461.1	PF01822.22	na	peroxidase	lignin
AA3	KKA28521.1	PF16010.8	1.1.99.18	Cellobiose dehydrogenase	cellulose
	KKA30608.1	PF16010.8	1.1.99.18	Cellobiose dehydrogenase	cellulose
	KKA31082.1	PF00732.22	1.1.3.7	aryl alcohol oxidase	lignin
AA5	KKA28638.1	PF01822.22	1.1.3.-	Oxidase with oxygen as acceptor	lignin
AA7	KKA27659.1	PF01565.26	1.1.3.-	glucooligosaccharide oxidase	cellobiose
	KKA30937.1	PF01565.26	1.1.3.-	glucooligosaccharide oxidase	cellobiose
AA9	KKA27328.1	PF03443.17	1.14.99.54	lytic cellulose monooxygenase	cellulose
	KKA28212.1	PF03443.17	1.14.99.54	lytic cellulose monooxygenase	cellulose
	KKA28497.1	PF03443.17	1.14.99.54	lytic cellulose monooxygenase	cellulose
	KKA25992.1	PF03443.17	1.14.99.54	endo-β-1,4-glucanase	cellulose
	KKA25994.1	PF03443.17	1.14.99.54	endo-β-1,4-glucanase	cellulose
	KKA29038.1	PF03443.17	1.14.99.54	lytic cellulose monooxygenase	cellulose
	KKA29219.1	PF03443.17	1.14.99.54	lytic cellulose monooxygenase	cellulose
CE4	KKA26186.1	PF01522.24	3.5.1.41	chitin deacetylase	chitin
	KKA27343.1	PF01522.24	3.5.1.41	chitin deacetylase	chitin
CE5	KKA30377.1	PF01083.25	3.1.1.72	acetyl xylan esterase	Hemi cellulose (xylan)
	KKA30382.1	PF01083.25	3.1.1.72	acetyl xylan esterase	Hemi cellulose (xylan)
GH03	KKA26832.1	PF01915.25	3.2.1.21	β-glucosidase	Cellulose, Hemi cellulose
	KKA30767.1	PF01915.25	3.2.1.21	β-glucosidase	Cellulose, Hemi cellulose
GH05	KKA26007.1	PF00150.21	3.2.1.4	endo-β-1,4-glucanase	cellulose
	KKA26778.1	PF00150.21	3.2.1.4	endo-β-1,4-glucanase	cellulose
	KKA28137.1	PF00150.21	3.2.1.4	endo-β-1,4-glucanase	cellulose
GH07	KKA26295.1	PF00840.23	3.2.1.176	cellulose 1,4-beta-cellobiosidase	cellulose
	KKA28489.1	PF00840.23	3.2.1.4	endo-β-1,4-glucanase	cellulose
GH10	KKA27891.1	PF00331.23	3.2.1.8	endo-1,4-β-xylanase	Hemi cellulose (xylan)
	KKA29568.1	PF00331.23	3.2.1.8	endo-1,4-β-xylanase	Hemi cellulose (xylan)
GH11	KKA29107.1	PF00457.20	3.2.1.8	endo-β-1,4-xylanase	Hemi cellulose (xylan)
	KKA30007.1	PF00457.20	3.2.1.8	endo-β-1,4-xylanase	Hemi cellulose (xylan)
GH115	KKA28239.1	PF15979.8	3.2.1.131	xylan α-1,2-glucuronidase	Hemi cellulose (xylan)
GH125	KKA28305.1	PF06824.14	-	exo-α-1,6-mannosidase	mannan (Hemi cellulose)
GH128	KKA29105.1	PF11790.11	-	β-1,3-glucanase	β-glucans
GH13	KKA30803.1	PF00128.27	3.2.1.1	α-amylase	starch
GH131	KKA28951.1	PF18271.4	3.2.1.21	endo-β-1,4-glucanase	starch
	KKA29646.1	PF18271.4	3.2.1.21	endo-β-1,4-glucanase	starch
GH132	KKA26122.1	PF03856.16	3.2.1.-	Beta-glucosidase	starch
GH15	KKA29558.1	PF00723.24	3.2.1.3	glucoamylase	starch
GH16	KKA26151.1	PF00722.24	3.2.1.73	licheninase	Starch
	KKA27451.1	PF00722.24	3.2.1.73	licheninase	Starch
	KKA28499.1	PF00722.24	3.2.1.73	licheninase	Starch
	KKA30944.1	PF00722.24	3.2.1.181	endo-β-1,3-galactanase	Pectin (Arabinogalactan)
GH18	KKA26416.1	PF03009.20	N		Polysaccharides
	KKA27515.1	PF00704.31	3.2.1.14	chitinase	chitin
	KKA30054.1	PF00704.31	3.2.1.14	chitinase	chitin
	KKA30697.1	PF00704.31	3.2.1.14	chitinase	chitin
GH20	KKA30299.1	PF00728.25	3.2.1.52	β-hexosaminidase	Polysaccharides
GH28	KKA31208.1	PF00295.20	3.2.1.15	polygalacturonase	Pectin
GH30	KKA27339.1	PF14587.9	3.2.1.164	endo-β-1,6-galactanase	Pectin (Arabinogalactan)
	KKA29858.1	PF02057.18	3.2.1.164	endo-β-1,6-galactanase	Pectin (Arabinogalactan)
GH32	KKA28220.1	PF00251.23	3.2.1.26	invertase	sucrose
GH37	KKA30799.1	PF01204.21	3.2.1.28	α,α-trehalase	Trehalose
GH38	KKA26248.1	PF01532.23	3.2.1.24	α-mannosidase	Hemi cellulose (mannan)
GH43	KKA27803.1	PF04616.17	3.2.1.99	endo-α-1,5-L-arabinanase	Hemi cellulose (xylan)
	KKA28970.1	PF04616.17	3.2.1.145	exo-β-1,3-galactanase	Hemi cellulose (xylan)
	KKA29859.1	PF04616.17	3.2.1.145	exo-β-1,3-galactanase	Hemi cellulose (xylan)
	KKA30545.1	PF04616.17	3.2.1.145	exo-β-1,3-galactanase	Hemi cellulose (xylan)
GH45	KKA28018.1	PF02015.19	3.2.1.4	endo-β-1,4-glucanase	cellulose
GH51	KKA29147.1	PF06964.15	3.2.1.55	α-L-arabinofuranosidase	Hemicellulose
GH53	KKA29651.1	PF07745.16	3.2.1.89	endo-β-1,4-galactanase	Hemicellulose
GH55	KKA30026.1	PF12708.10	3.2.1.58	glucan β-1,3-glucosidase	callose
	KKA30509.1	PF12708.10	3.2.1.58	glucan β-1,3-glucosidase	callose
GH64	KKA27366.1	PF16483.8	3.2.1.39	glucan endo-1,3-β-D-glucosidase	callose
GH76	KKA26192.1	PF03663.17	3.2.1.101	α-1,6-mannanase	mannan
GH78	KKA27604.1	PF17390.5	3.2.1.40	α-L-rhamnosidase	mannan
GH93	KKA30496.1	PF06964.15	3.2.1.-	exo-α-L-1,5-arabinanase	Hemicellulose
GT4	KKA26489.1	PF04488.18	2.4.1.257	α-1,6-mannosyltransferase	mannan
PL1	KKA26877.1	PF00544.22	4.2.2.10	pectin lyase	pectin
	KKA27238.1	PF00544.22	4.2.2.10	pectin lyase	pectin
PL3	KKA30830.1	PF03211.16	4.2.2.2	pectate lyase	pectin
PL4	KKA28462.1	PF09284.13	4.2.2.23	rhamnogalacturonan endolyase	pectin

**Table 3 jof-09-00303-t003:** Comparison of CAZymes in the secretome of six *Thielaviopsis* species.

CAZyme Class	Class Members	*Thielaviopsis* Species					
		*T. punctulata*	*T. ethacetica*	*T. cerberus*	*T. euricoi*	*T. musarum*	*T. populi*
Auxiliary Activities	AA1	3	1	1	2	1	2
	AA2	2	0	0	1	0	0
	AA3	3	3	0	3	2	1
	AA5	1	1	1	1	1	1
	AA7	2	0	0	0	2	0
	AA8	0	1	1	1		2
	AA9	7	5	0	5	5	3
	AA11	0	2	1	2	2	1
	AA12	0	1	0	0	0	0
	AA16	1	0	0	0	0	0
Carbohydrate esterases	CE1	0	1	1	2	2	1
	CE3	0	1	0	0	0	0
	CE4	2	1	1	1	0	0
	CE5	2	1	2	1	1	0
Glycoside Hydrolases	GH03	2	2	2	2	2	
	GH05	3	0	0	0	1	1
	GH06	0	0	0	1	0	0
	GH07	2	1	1	0	1	0
	GH10	2	1	1	2	2	1
	GH11	2	3	3	3	2	1
	GH12	0	0	1	0	0	0
	GH13	0	0	0	0	0	1
	GH15	1	0	0	0	0	1
	GH16	4	5	5	5	3	3
	GH17	4	3	4	3	4	4
	GH20	1	1	1	1	1	1
	GH28	1	0	1	0	0	0
	GH30	2	2	1	1	1	0
	GH31	0	0	0	0	1	0
	GH32	1	1	1	1	1	1
	GH37	1	0	1	0	0	0
	GH38	1	0	0	0	0	0
	GH43	4	4	4	5	3	2
	GH45	1	0	0	0	0	0
	GH51	1	0	0	0	0	0
	GH53	1	0	0	0	0	1
	GH55	2	1	1	1	0	1
	GH63	0	1	0	1	1	0
	GH64	1	0	0	0	0	0
	GH72	0	1	2	2	2	3
	GH74	0	0	1	1	0	0
	GH76	1	2	2	2	2	2
	GH78	1	0	1	0	0	0
	GH92	0	1	0	1	1	1
	GH93	1	1	0	0	1	0
	GH115	1	0	0	0	0	0
	GH125	1	0	0	0	0	0
	GH128	1	0	0	0	0	0
	GH131	2	0	0	0	0	1
	GH132	1	0	0	0	0	0
Glycosyl Transferases	GT4	1	0	0	0	0	0
	GT8	0	0	0	1	0	1
	GT32	0	0	0	0	0	1
	GT34	0	1	0	1	1	0
	GT61	0	1	0	0	0	0
Polysaccharide Lyases	PL1	2	2	1	1	1	2
	PL3	1	1	0	1	0	1
	PL4	1	0	0	0	0	1

**Table 4 jof-09-00303-t004:** Putative effectors in the secretome of *T. punctulata*.

Protein Id	Length	Cysteines	Domain1	Name	Domain2	Name
KKA28270.1	275	4	PF13640	2OG-FeII_Oxy_3		
KKA29775.1	392	4	PF12697	Abhydrolase_6		
KKA30577.1	359	7	PF00733	Asn_synthase		
KKA26543.1	316	5	PF00967	Barwin		
KKA27620.1	348	4	PF00188	CAP		
KKA28836.1	133	4	PF07249	Cerato-platanin		
KKA26938.1	356	6	PF00127	Copper-bind		
KKA28390.1	177	2	PF00127	Copper-bind		
KKA28890.1	234	4	PF00190	Cupin_1		
KKA27979.1	369	3	PF00775	Dioxygenase_C		
KKA28872.1	203	4	PF07510	DUF1524		
KKA26687.1	278	4	PF10057	DUF2294	PF13640	2OG-FeII_Oxy_3
KKA27329.1	237	3	PF10901	DUF2690	PF07883	Cupin_2
KKA26724.1	253	4	PF11693	DUF2990	PF11937	DUF3455
KKA26892.1	247	3	PF11937	DUF3455		
KKA27739.1	118	7	PF15371	DUF4599		
KKA29708.1	260	7	PF05359	DUF748		
KKA31236.1	179	4	PF02221	E1_DerP2_DerF2		
KKA26926.1	345	6	PF11327	Egh16-like	PF09716	ETRAMP
KKA26947.1	171	6	PF11327	Egh16-like		
KKA27553.1	158	4	PF11327	Egh16-like		
KKA27672.1	272	4	PF11327	Egh16-like		
KKA29410.1	381	8	PF11327	Egh16-like	PF12230	PRP21_like_P
KKA25960.1	307	2	PF03372	Exo_endo_phos		
KKA29592.1	316	4	PF13668	Ferritin_2	PF06140	Ifi-6-16
KKA29712.1	289	3	PF07554	FIVAR	PF00775	Dioxygenase_C
KKA26457.1	200	2	PF00254	FKBP_C		
KKA27476.1	255	2	PF05630	NPP1		
KKA29596.1	296	2	PF02585	PIG-L		
KKA28719.1	290	8	PF16670	PI-PLC-C1		
KKA28033.1	266	2	PF13883	Pyrid_oxidase_2		
KKA29951.1	304	5	PF02265	S1-P1_nuclease		
KKA28667.1	316	5	PF12138	Spherulin4	PF15862	Coilin_N
KKA29465.1	127	4	PF14558	TRP_N		
KKA26744.1	291	2	No domain			
KKA27434.1	157	5	No domain			
KKA27537.1	98	6	No domain			
KKA27913.1	179	4	No domain			
KKA28484.1	339	8	No domain			
KKA29066.1	211	6	No domain			
KKA29758.1	78	7	No domain			
KKA30719.1	189	2	No domain			
KKA30870.1	169	4	No domain			
KKA30907.1	140	2	No domain			
KKA30938.1	371	8	No domain			
KKA31074.1	140	2	No domain			
KKA31087.1	113	8	No domain			

## Data Availability

Not Applicable.

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
