# Peer review of "In Silico Characterization of the Secretome of the Fungal Pathogen Thielaviopsis punctulata, the Causal Agent of Date Palm Black Scorch Disease"

_jof, 2023, doi:10.3390/jof9030303_

Round 1

Reviewer 1 Report

The manuscript about the prediction of potential virulence factors is interesting and important due to the possibility of understanding the processes associated with fungal infections of plants. Their identification with the in silico method should accelerate further protein identification and functional tests, and thus translate into the search for effective prevention methods.

I have some comments on the manuscript. The first is numerous punctuation and spelling errors. An example is the word parthogen in the title. There are many such errors in the manuscript and it requires careful reading and correction throughout the text. Similarly, species names should be in italics, and in many places they are not.

Figure 2, panels B,C,D,E - captions of the Y axis are missing, numbers on the X axis are not clearly visible, no captions of the X axis, panel C,D,E.

Figure 3A, 4A, 5 - caption of the Y axis is missing.

Figure 6 and its caption should be better explained.

Author Response

Dear Reviewer,

Thanks for your valuable comments and suggestion. Please see my response in the attached pdf file

Thanks

Reviewer 2 Report

Based on the known draft genome sequences, the authors carried out a bioinformatic analysis of the secretomes of the fungal pathogen Thielaviopsis punctulata, which cause great damage to the date palm crop, and five other Thielaviopsis species. It should be noted in the manuscript are the compared species also pathogens, parasitizing on palms, as well as on bananas and cocoa trees. If so, the following title may be more appropriate for the topic of the article: “In silico characterization of the secretomes of Thielaviopsis punctulata, the causal agent of date palm black scorch disease and some other phytopathogens of the same genera”.

When discussing the results, it should be noted not only the secretome specific for T. punctulata, but also secretome common features for all compared species.

 Specify the species used in the construction of the tree. Compare this tree with the results obtained from the analysis of DNA markers (in particular, Alhudaib et al., Plants 2022, 11, 250. https://doi.org/10.3390/plants11030250 )

 There are also a few technical notes.

In Figure 1, the spelling of the species Thielaviopsis punctulata is incorrect

The method of constructing an evolutionary tree should be described not in the legend to Figure 1, but in the Materials and Methods.

 Lines 12,20-23, 133, 176, 199,228,322, 369-370, 390-391, 440-443, 446, 469, 492: change fonts of species names to italic

 Line 128: specie > species

Line 391: punctulat > punctulata

Suppl.Table 1: titles of columns are absent

 Suppl.Table 3: is entitled as Table S2 by a mistake. Change fonts of species names to italic. Describe meaning of colored boxes

 The GT class is mentioned on line 240, but not in Figure 4. In the legend of Figure 4A, the abbreviations need to be deciphered. Name the y-axis in Figure 4A. Change fonts of species names to italic.

Author Response

Dear Reviewer,

Thanks for your valuable comments and suggestions. Please find my response in the attached pdf file

Thanks

Reviewer 3 Report

Through bioinformatics, the authors interrogate the secretome of the plant fungal pathogen Thielaviopsis punctulata. A number of proteins were identified that are likely to contribute to the virulence of the fungus.

Overall, the work is sound and will be of interest to readers of the journal. I recommend citing other recent studies and placing their results in context, such as the following-

10.1371/journal.pone.0260830

10.1186/s12864-021-07902-w

Author Response

(The authors gave the same response as above.)
